# Mediterranean Pine Vole, *Microtus duodecimcostatus:* A Paradigm of an Opportunistic Breeder

**DOI:** 10.3390/ani11061639

**Published:** 2021-06-01

**Authors:** Miguel Lao-Pérez, Diaa Massoud, Francisca M. Real, Alicia Hurtado, Esperanza Ortega, Miguel Burgos, Rafael Jiménez, Francisco J. Barrionuevo

**Affiliations:** 1Departamento de Genética e Instituto de Biotecnología, Lab. 127, Centro de Investigación Biomédica, Universidad de Granada, Avenida del Conocimiento S/N, 18016 Armilla, Granada, Spain; mlao@ugr.es (M.L.-P.); dfm00@fayoum.edu.eg (D.M.); martinez@molgen.mpg.de (F.M.R.); alhurtado@ugr.es (A.H.); mburgos@go.ugr.es (M.B.); 2Departamento de Bioquímica y Biología Molecular III e Inmunología, Facultad de Medicina, Universidad de Granada, 18016 Armilla, Granada, Spain; esortega@ugr.es

**Keywords:** seasonal breeding, testis regression, *Microtus duodecimcostatus*, opportunistic breeding

## Abstract

**Simple Summary:**

In temperate zones of the Earth, some mammalian species reproduce seasonally whereas others do it continuously. Other species are summer breeders in the north and winter breeders in the south. Thus, the reproductive pattern seems not to be a species-specific but a population-specific trait. We investigated the reproduction pattern of the Mediterranean pine vole, *Microtus duodecimcostatus*, in the area around the city of Granada in Southern Spain, and found that individuals living in wastelands reproduce seasonally whereas those living in close poplar plantations (just 8 km apart) reproduce throughout the year, as did voles captured in wastelands and kept in captivity. These animals represent thus a paradigm of an opportunistic breeder as particular individuals stop breeding or not, depending on the environmental conditions they face at any moment. Sexually inactive male voles undergo complete testis inactivation and their sperm production is halted. The immune system in active testes is depressed, a phenomenon known as “immune privilege” that protect germ cells from autoimmune attack. We studied gene activity in active and inactive testes and our results indicate that such an immune privilege is lost in inactive testes, suggesting an important role for this process during testis regression.

**Abstract:**

Most mammalian species of the temperate zones of the Earth reproduce seasonally, existing a non-breeding period in which the gonads of both sexes undergo functional regression. It is widely accepted that photoperiod is the principal environmental cue controlling these seasonal changes, although several exceptions have been described in other mammalian species in which breeding depends on cues such as food or water availability. We studied the circannual reproductive cycle in males of the Mediterranean pine vole, *Microtus duodecimcostatus*, in the Southeastern Iberian Peninsula. Morphological, hormonal, functional, molecular and transcriptomic analyses were performed. As reported for populations of other species from the same geographic area, male voles captured in wastelands underwent seasonal testis regression in summer whereas, surprisingly, those living either in close poplar plantations or in our animal house reproduced throughout the year, showing that it is the microenvironment of a particular vole subpopulation what determines its reproductive status and that these animals are pure opportunistic, photoperiod-independent breeders. In addition, we show that several molecular pathways, including MAPK, are deregulated and that the testicular “immune privilege” is lost in the inactive testes, providing novel mechanisms linking seasonal testosterone reduction and testis regression.

## 1. Introduction

The existence of climatic seasons force species of the temperate zones of the Earth to follow a circannual biological cycle in which reproduction and other functions concentrate in those periods with more favorable conditions. Since individuals do not need their gametes during the non-breeding season, both males and females undergo significant changes in their reproductive systems in this period [1]. In many species, females experience a period of anoestrus and reduced or null sexual receptivity, whereas males undergo a process of testis regression by which they lose most of the germinative epithelium of their testes, with a considerable reduction of the testicular mass and volume [2].

The mechanisms of testis regression have been studied in many species of vertebrates, including reptiles, amphibians, birds and mammals. Apoptosis is the main testis-regression effector in birds, reptiles and amphibians [3,4,5,6,7,8]. However, an alternative mechanism has been described in a mammalian species, the Iberian mole, *Talpa occidentalis* [9,10], in which the main testis-involution effector is desquamation of living, non-apoptotic germ cells. This process is regulated by modulating the expression and distribution of cell-adhesion molecules in the seminiferous epithelium. The same mechanism has been reported also in other mammals, including the long hairy armadillo, *Chaetophractus villosus* [11,12], and the Egyptian long-eared hedgehog, *Hemiechinus auritus* [13].

It has long been known that reproduction is controlled by the hypothalamic–pituitary–gonadal (HPG) axis, where kisspeptin neuropeptides, gonadotrophin-releasing hormone (GnRH), pituitary gonadotrophins (luteinizing hormone (LH) and follicle-stimulating hormone (FSH)) and gonadal steroid hormones are the most important activating elements [14]. There are several environmental cues that may control seasonal reproduction but photoperiod is by far the best known. In mammals, the phototransduction pathway, which includes the retina, transduces the photic information via the retinohypothalamic tract to nuclei in the hypothalamus and finally to the pineal gland, which produces melatonin at night that exerts an inhibitory effect on the reproductive system [15]. However, other factors, such as food and water availability, stress and weather, can either modify or even overcome the influence of photoperiod. Environmental cues modulate the function of the HPG axis. The levels of serum gonadotropins are lower in the non-breeding period, with the subsequent reduction of circulating testosterone, which leads to spermatogenesis inhibition [16]. Photoperiod is considered the most common environmental cue controlling seasonal reproduction, although the number of known seasonal breeders, which are photoperiod-independent is growing. However, the alternative non-photic effector cue could be identified in very few species, including the Californian mouse, *Peromyscus californicus*, in which reproduction depends on water availability [17], the Merino sheep, *Ovis aries*, where male fertility depends on food availability and social factors [18] and the musk shrew, *Suncus murinus*, a species in which females completely lose sexual receptivity when nutrition is insufficient [19]. The seasonal breeding pattern of different populations of the same species can also vary with latitude, as described for instance in the greater white-toothed shrew, *Crocidura russula*, in the Iberian peninsula, where northern populations stop breeding in winter whereas southern ones do it in summer [20,21]. This fact suggests that a particular pattern of seasonal breeding is attributable to particular populations of a given species, but not to the species itself [2]. Moreover, some vertebrates have an opportunistic breeding strategy which implies that reproduction takes place as long as the environmental conditions remain favorable, independently of photoperiod, latitude or altitude, as shown in many rodent species [22].

The Mediterranean pine vole, *Microtus duodecimcostatus*, is distributed across most of the Iberian Peninsula, except in the north-western region, and in Southern France [23]. It is characterized by a burrowing behavior spending most time in underground tunnels and burrows. In the studied areas, this vole can be considered sexually active throughout the year, with a peak of sexual activity in winter and spring, from November to May with a maximum in February/March [24]. In the province of Granada, this species is relatively abundant in various habitats including cereal crops, almond tree plantations, wastelands, poplar groves and irrigated vegetable gardens, where it causes occasional pest damages with a pluriannual frequency (our personal observations).

The reproduction timing of this species in this geographic area has not been studied to date and this is the aim of this work. We report here the results of a comprehensive study of the testes in different populations of *M. duodecimcostatus* living at different habitats throughout the circannual cycle of reproduction in Southeastern Iberian Peninsula. Histological and immunohistological studies, including morphometry, blood–testis barrier (BTB) permeability tests, serum testosterone level determinations, apoptotic cell detection techniques and transcriptomic analyses were performed. Our results show that the analyzed voles are pure opportunistic breeders and suggest that in regressed testes several molecular pathways involved the control of spermatogenesis and BTB dynamics are deregulated and the intensity of the immune response is altered.

## 2. Materials and Methods

### 2.1. Animals

Forty six Mediterranean pine vole adult males were initially captured alive in isolated small wastelands populations near the localities of Las Gabias and La Malaha (Granada province, Southeastern Spain) throughout two consecutive years. Four study groups were established according to the season of capture (winter, *n* = 19; spring, *n* = 9; summer, *n* = 6; autumn, *n* = 12; Appendix A). We also captured four males in poplar groves in the locality of Santa Fe (Granada province, Southeastern Spain), just 9 km apart from the wastelands, during the summer (Appendix A). We used subterranean one-way, one-door, cylindric wire-mesh traps designed and made in our laboratory. Traps were baited with pieces of apple, carrot and potato. In cold months (late autumn, winter and early spring) traps were set at daytime whereas in the warm months (late spring, summer and early autumn) captures were done at night. Animals were captured with the permission of the Andalusian environmental authorities (Consejería de Agricultura, Pesca y Medio Ambiente) and handled following the guidelines and approval of both the Ethical Committee for Animal Experimentation of the University of Granada and the Andalusian Council of Agriculture and Fisheries and Rural Development (Registration number: 450-19131; 16 June 2014). In this study, we also included unpublished morphometric data and paraffin-embedded testes from males of the laboratory colony of *Pitymys duodecimcostatus* (the former name of the Mediterranean pine vole) we maintained during the entire year of 1990 for a different study. We analyzed testes from 13 males euthanised in July to show that these animals do not stop breeding during the summer in our animal house conditions (water and food ad libitum (carrot, apple and mouse pellets)), temperature around 20–25 degrees and natural day light.

### 2.2. Histology and Immunohistological Methods

Animals were transported to our laboratory and euthanised by CO_2_ inhalation the day of capture, dissected and the testes were removed, weighed and fixed overnight in a 50× volume of Serra’s fixative, a mixture of 100% ethanol, 40% formaldehyde and glacial acetic acid in proportions of 60:30:10, respectively. Epididymides were processed in the same way for further histological studies. After fixation, testes and epididymides were dehydrated and embedded in paraffin, sectioned at 5 μm, mounted on polylysine-coated slides (VWR, Leuven, Belgium), and stained with hematoxylin and eosin according to standard methodology. For immunohistochemistry, testis sections were deparaffinized and mounted on slides that were processed using the ABC Kit (Vector Laboratories, Burlingame, CA, USA), according to the manufacturer’s instructions. For single and double immunofluorescence, testis sections were incubated with primary antibodies overnight, washed, incubated with suitable conjugated secondary antibodies at room temperature for 1 h and counter-stained with 4′,6-diamino-2-phenylindol (DAPI). We used a Nikon Eclipse Ti microscope equipped with a Nikon DS-Fi1c digital camera (Nikon Corporation, Tokyo, Japan) to take photomicrographs. In negative controls, the primary antibody was omitted. Appendix A summarizes the antibodies and working concentrations used in this study.

### 2.3. Cell Death Analysis

Apoptotic cells were detected in testis histological sections by using the TUNEL (terminal deoxynucleotidyl transferase dUTP nick end labeling) technique with the fluorescent in situ cell death detection kit (Roche, Mannheim, Germany), according to the manufacturer’s guidelines. The enzyme solution was omitted to prepare negative controls.

### 2.4. Serum Testosterone Levels

Blood samples were obtained from male voles by heart puncture, stored at 4 °C overnight and centrifuged at 6000 rpm for 20 min at 4 °C. The supernatant was isolated and stored at −80 °C until used. Testosterone concentrations were determined by radioimmunoassay (RIA) with the DRG testosterone RIA (CT) kit (DRG, New York, NY, USA), according to standard procedures. The analytical sensitivity of the kit was 0.05 ng/mL, the intra-assay coefficient of variation was 3.3% and the interassay coefficient of variation was 4.8%. Duplicate measurements were made for all animals. The specificity (percentage of cross-reaction estimated by comparing the concentration yielding a 50% inhibition) was 0.28% for androstenedione and 0.31% for dihydrotestosterone.

### 2.5. BTB Permeability

To check the permeability of the BTB, both sexually active and inactive males of *M. duodecimcostatus* (two of each) were anaesthetized with 0.125% avertin (2,2,2-tribromoethanol), their testes were exposed and a total of 50 μL of 10 mg/mL of EZ-Link Sulfo-NHS-LC-Biotin (Pierce Chemical Co., Rockford, IL, USA) diluted in PBS containing 1 mM of CaCl_2_ were injected beneath the tunica albuginea of the left testis by several punctures. As a control, the right testis was injected in a similar way with PBS containing 1 mM of CaCl2. The testes were placed again inside the body, and the voles were euthanised after 30 min. The testes were immediately collected, fixed overnight in 4% paraformaldehyde and paraffin-embedded following standard methods. Sections (7 μm thick) were deparaffinized, rehydrated and incubated for 30 min with an Alexa Fluor 568-conjugated streptavidin solution (included in the tracer kit) at 25 °C and counterstained with DAPI.

### 2.6. Transcriptome Analysis

Total RNA was isolated from both testes of three males captured in the wastelands during the summer (reproductively inactive) and three males captured during the winter (reproductively active) using the Qiagen RNeasy Midi kit following the manufacturer’s instructions. After successfully passing quality check, the RNAs samples were paired-end sequenced separately in an Illumina HiSeq 2000 platform at the Max Planck Institute for Molecular Genetics facilities in Berlin and the quality of the resulting sequencing reads was assessed using FastQC (http://www.bioinformatics.bbsrc.ac.uk/projects/fastqc/). A detailed protocol of the bioinformatic tools used to analyze transcriptomic data is provided in Appendix A. Briefly, For transcriptome assembly we used the Oyster River Protocol (ORP) [25] and for transcriptome annotation we used the trinotate pipeline [26], with some modifications. For transcriptome normalization, we used the previously published single cell RNA-seq data [27,28] to remove germ cell-specific transcripts. For transcript quantification, we used Trinity with the ORP-assembled transcriptome as a reference [29,30], and for differential expression analysis we used the Trinity/RSEM package. Genes were considered as differentially expressed at p.adjust < 0.001 and logFC > 2. Gene Ontology analysis was performed with GOATOOLS [31] and with the enrichGO function of the clusterProfiler bioconductor package [32]. General terms and terms not related with testicular functions were not displayed. For visualization, we used the barplot function of the same suite, and the bioconductor pathview package [33]. For the gene-concept analysis the cnetplot function of the clusterProfiler package was used.

## 3. Results

### 3.1. Males of M. duodecimcostatus from the South-Eastern Iberian Peninsula Undergo Summer Testis Regression in Wastelands but Not in Other Habitats

To study the reproductive cycle of *M. duodecimcostatus* in the south-east of the Iberian Peninsula we captured males in wastelands throughout the year and four study groups were established according to the season of capture. Data on body mass, testis mass and relative mass of the testis (testis mass (mg)/body mass (mg) × 100) together with the matrix of *p* values corresponding to the statistical Student’s *t*-tests are summarized in Appendix A. The body mass showed no statistically significant differences between groups, except in the comparison between males from winter and spring, although these differences were small and almost not significant (*p* = 0.48; Figure 1a; Appendix A). Contrarily, statistical analysis revealed a highly significant reduction of the testis mass and the relative testis mass in the summer group when compared with the other three groups, showing the existence of summer testis regression in this species (Figure 1b,c; Appendix A). Accordingly, the histological analysis showed that, whereas males captured in non-summer seasons had large, spermatogenically active testes and epididymides full of spermatozoa (Figure 1g,k; only winter and summer testis sections are shown), males captured in summer had regressed testes with reduced seminiferous tubules and empty epididymides (Figure 1h,l). Later, we also found small populations of Mediterranean pine voles in the extensive poplar cultivations near the locality of Santa Fe, only 9 km apart from the wastelands locations where we performed our initial study. Since the environmental conditions in wastelands and in poplar groves are quite different (see below), we decided to capture some few males in these poplar groves during the summer to check whether they also experience testis regression. Contrarily, we found that they were reproductively active, showing no testicular involution. In addition, we also checked our files of previous studies carried out in our laboratory with this species during 1990–1991 [34], when we maintained for almost one entire year a colony of Mediterranean pine voles exposed to natural photoperiod in our animal house facilities. These voles bred continuously, even in the summer and, fortunately, we kept morphometric data and histological testicular material from some of these individuals. Appendix A summarizes the body mass, testis mass and relative mass of the testis, together with the matrix of *p* values corresponding to the statistical Student’s *t*-tests of voles captured in the wastelands during summer and winter, and males from both the poplar groves and laboratory animal facilities during the summer. Body mass of males captured in the wastelands were around 20% smaller than those from either the poplar groves or the animal house facilities, as expected if we consider the harder environmental conditions of this habitat (Figure 1d; Appendix A). Only the testes of males captured in the wastelands during the summer showed a significant mass reduction when compared to those of the other groups (Figure 1e; Appendix A). Similar results were obtained regarding the relative testis mass (Figure 1f; Appendix A). According to these data, the histological analysis showed that summer voles from both poplar groves and animal house facility had testes with fully active seminiferous tubules and epididymes containing abundant spermatozoa (Figure 1i,j,m,n). In summary, only males living in the wastelands undergo testis regression during the summer, whereas the rest keep breeding throughout the year.

### 3.2. Spermatogonenesis Is Arrested at the Spermatogonial Stage in Regressed Testes of M. duodecimcostatus

Next, we focused on the analysis of active and regressed testes of Mediterranean pine voles captured in the wastelands. Immunohistochemistry for SOX9, a marker for Sertoli cells [35], revealed that active testes show a spermatogenic stage-dependent expression pattern, being stronger in stages I-IV and weaker in stages VII-IX (Figure 2A), similar to that reported previously for both the rat and the Iberian mole [36,37]. In contrast, we observed uniform SOX9 immunoreactivity in regressed testes, where the distance between neighboring SOX9-positive cells was reduced to a minimum, the nuclei almost touching each other in most cases (Figure 2B). This shows strong shrinkage affecting both seminiferous tubules and Sertoli cell cytoplasm in these regressed testes. We also studied the expression of DMC1, a marker for zygotene and early pachytene spermatocytes [38]. Immunofluorescence revealed a spermatogenic cycle-dependent expression of DMC1 in active testes, with DMC1-positive cells present only in stages VII-IX (Figure 2C). In contrast, almost all seminiferous tubules were devoid of DMC1 positive cells in the regressed testes (Figure 2D), showing that spermatogonial cells do not enter meiosis in this inactive period. Double immunofluorescence for SOX9 and DMRT1, which is expressed in both Sertoli and spermatogonial cells [39] showed that the seminiferous tubules of active testes contained SOX9^+^, DMRT1^+^ Sertoli cells (Figure 2E,G,I; arrowheads) and SOX9^−^, DMRT1^+^ spermatogonial cells, both located in basal positions of the seminiferous tubules (Figure 2E,G,I; arrows). We also observed many SOX9^−^, DMRT1^−^ cells inside the tubules, corresponding to germ cells in more advanced stages of the spermatogenic cycle (Figure 2E,G,I). In the regressed testes, seminiferous tubules contained SOX9^+^, DMRT1^+^ Sertoli cells, whose nuclei occupied the most inner positions of the tubules (Figure 2F,H,J; arrowheads), and SOX9^−^, DMRT1^+^ spermatogonial cells located always in basal positions (Figure 2F,H,J; arrows). No SOX9^−^, DMRT1^−^ cells were observed inside these tubules (Figure 2F,H,J). Altogether, these results indicate that the testes of sexually inactive males of Mediterranean pine vole males undergo a profound regression in which meiosis is abolished and thus spermatogenesis is completely halted at the spermatogonial stage.

### 3.3. The Blood Testis Barrier Is Impaired in Regressed Testis of M. duodecimcostatus

We checked the status of the BTB in both active and inactive testes of *M. duodecimcostatus* using immunofluorescence for CLAUDIN11, a principal component of the tight junctions forming the BTB [40]. We found a sharp CLAUDIN11 immunoreactivity in the basal areas of the seminiferous tubules in active testes (Figure 3A). In contrast, CLAUDIN11 expression was completely disorganized in the seminiferous tubules of regressed testes (Figure 3B). We also checked the integrity of the BTB by injecting a biotin tracer into the interstitial space of the testes. In active ones, we detected biotin in both the interstitial tissue and the basal compartment of the seminiferous tubules, but not in the adluminal compartment (Figure 3C). In contrast, the biotin tracer was clearly detected in both basal and adluminal compartments of the regressing testes. This is better observed in non-completely regressed testes like that shown in Figure 3D. These results show that the BTB is permeated in the regressed testes of sexually inactive males.

### 3.4. Reduced Steroidogenesis but Not Increased Testicular Apoptosis in Sexually Inactive Males of M. duodecimcostatus

We studied the androgenic function of both sexually active and inactive males living in the wastelands. Immunohistochemistry for the cholesterol side-chain cleavage enzyme (P450scc) revealed a strong expression in Leydig cells, with no apparent difference in the signal intensity between active and regressed testes (Figure 4A,B). The serum concentration of testosterone was determined by radioimmunoassay (RIA). We found a significant 80% reduction of the serum testosterone levels in sexually inactive males, compared to that of active ones (active: 0.13 ± 0.09 ng/mL, *n* = 10; inactive: 0.023 ± 0.18 ng/mL, *n* = 6; two-tailed *t*-test, *p* = 0,015; Figure 4C). We also measured the serum testosterone concentration of males from the poplar groves in the summer (active) (0.22 ± 0.08 ng/mL; *n* = 3), and found that it was 10 times higher than that of males from the wastelands at the same season (inactive) (two-tailed *t*-test, *p* = 5.7 × 10^−4^; Figure 4C), but not significantly higher than that of males from the wastelands in winter (active) (two-tailed *t*-test, *p* = 0.18; Figure 4C). Since reduced levels of testosterone is associated with increased testicular apoptosis [41], we checked the incidence of cell death in both active and inactive testes of *M. duodecimcostatus* males from the wastelands using the TUNEL assays. In both groups, most of the seminiferous tubules were devoid of apoptotic cells (Figure 4D,E) and differences in the number of apoptotic cells between active and inactive testes were not statistically significant (active: 5.6 ± 2.3 apoptotic cells/0.1 mm^2^ of testis section; inactive: 4.0 ± 2.3 cells/0.1 mm^2^; Wilcox test, *p* = 0.066; Figure 4F).

### 3.5. Transcriptomic Differences between Active and Regressed Testes of M. duodecimcostatus

To find differences in gene expression, we performed RNA-seq on three active and three regressed testes from males captured in the wastelands. Now we know that regressed testes show spermatogenic arrest and, therefore, the higher expression levels of both meiotic and postmeiotic genes that we would detect in the active testes when compared with the regressed ones would not reflect physiological changes in gene expression but rather differences in germ cell content between both types of testes. This over-representation of germ-cell specific markers in sexually active testes would mask the differential gene expression produced in somatic cells, which are responsible for the regulation of spermatogenesis, in particular that in Sertoli cells. Due to this, we decided to exclude a number of known germ cell-specific markers in our transcriptomic analysis, as explained in Appendix A. When doing this, the mean correlation coefficient between samples from different category increased from 0.5 to 0.7 after normalization, confirming that our approach removed differences derived from the distinct germ cell content present in active and inactive testes. We found 3809 differentially expressed genes (DEG), from which 2623 were upregulated in active testes (Act-up-DEG) and 1186 upregulated in inactive testes (Inact-up-DEG) (FDR < 0.001 and |logFC| > 2; Figure 5a; Appendix A). Note that in Act-up-DEG still are many genes expressed in meiotic and postmeiotic germ cells, which were not included in the list of germ cell-specific genes that we removed in our analysis. Gene Ontology (GO) analysis of DEG, including both Act-up-DEG and Inact-up-DEG, showed a significant enrichment (p.adjust < 0.05) in a number of categories, many of them closely related and associated with different biological functions (Figure 5b; Appendix A). Within the most significant ones, we found several related categories involved in spermatogenesis and spermiogenesis (Figure 5b) such as spermatogenesis (GO:0007283; p.adjust = 1.41 × 10^−5^), male gamete differentiation (GO:0048232; p.adjust = 1.19 × 10^−5^) and sperm motility (GO:0097722; p.adjust = 1.44 × 10^−5^), among others. These categories were enriched in Act-up-DEG (Figure 5c; Appendix A), and were also found in the GO analysis of Act-up-DEG but not in that of Inact-up-DEG (Appendix A). Most of the genes in these categories corresponded to germ-cell markers, in which, as mentioned above, the differential expression is probably the result of different germ cell content of both types of gonads. However, we also found some Sertoli-specific Inact-up-DEG such as *Sox9*, *Sox8*, *Gata4* and *Wt1* (Appendix A and Appendix A). We also found terms enriched in genes associated with different testicular processes that may affect seasonal testis regression (Figure 5b) such as regulation of blood vessel size (GO:0050880; p.adjust = 1.4 × 10^−3^), angiogenesis (GO:0001525; p.adjust = 1.0 × 10^−2^), smooth muscle contraction (GO:0006939; p.adjust = 3.3 × 10^−3^), regulation of phagocytosis (GO:0060099; p.adjust = 1.4 × 10^−2^) and negative regulation of androgen receptor signaling pathway (GO:0060766; p.adjust = 3.1 × 10^−2^). All these categories were also found in the GO analysis of Inact-up-DEG but not in that of Act-up-DEG (Figure 5b; Appendix A). Sertoli cells play fundamental roles in supporting spermatogenesis including nutrient supply, maintenance of cell junctions and regulation of BTB dynamics. They also regulate the entry of germ cells into mitosis and meiosis. We showed that the BTB is disrupted in inactive testes and found a GO term enriched in genes involved in the negative regulation of cell–cell adhesion. We also found several GO terms related to Sertoli cell signaling involved in the regulation of spermatogenesis and the BTB dynamics (Figure 2D; Appendix A), including positive regulation of MAPK cascade (GO:0043410; p.adjust = 7.0 × 10^−3^) [42], ERK1 and ERK2 cascade (GO:0070371; p.adjust = 4.5 × 10^−5^; Zhang et al., 2014), phosphatidylinositol 3-kinase signaling (GO:0014065; p.adjust = 2.8 × 10^−2^) [43] and regulation of cytosolic calcium ion concentration (GO:0051480; p.adjust = 2.7 × 10^−4^) [44]. All these terms were also found in the GO analysis of Inact-up-DEG (Appendix A), among which we also found categories involved in the genetic control of the BTB integrity and spermatogenesis such as canonical WNT signaling pathway (GO:0060070; p.adjust = 1.8 × 10^−5^) [45] and cellular response to transforming growth factor beta stimulus (GO:0071560; p.adjust = 1.3 × 10^−10^) [42]. Gene-concept analysis using data from the GO analysis of all DEGs showed a very complex network in which the MAPK/ERK signaling pathway occupied the central region sharing many genes with the other categories (Figure 5d; Appendix A). Visualization of the DEG in the MAPK KEGG pathway (mmu04010) confirmed a higher abundance of Inact-up-DEG in all the steps of this signaling pathway (Appendix A). In this analysis, we also included the term “negative regulation of cell-cell adhesion” (GO:0022408; p.adjust = 4.5 × 10^−2^), which shared many genes with the other signaling pathways (Figure 5d). Altogether these results indicate that activation of the MAPK/ERK signaling pathway, and its interaction with other pathways such as WNT, TGF-β, regulation of cytosolic Ca2+ and PI3K plays important roles in the control of testicular functions during the breeding cycle of *M. duodecimcostatus*, including the regulation of cell adhesion molecules.

In the GO analysis of DEG we also identified several terms related to immune response (Figure 5b) such as macrophage activation (GO:0042116; p.adjust = 2.6 × 10^−3^), positive regulation of inflammatory response (GO:0050729; p.adjust = 4.0 × 10^−3^) and regulation of lymphocyte proliferation (GO:0050670; p.adjust = 2.1 × 10^−2^), among others. We also found terms related to pathways involved in the genetic control of the immune system such as regulation of tumor necrosis factor production (GO:0032680; p.adjust = 1.5 × 10^−2^) and regulation of cytokine secretion (GO:0050707; p.adjust = 1.8 × 10^−2^). As before, all these terms were found in the GO analysis of Inact-up-DEG but not in that of Act-up-DEG (Appendix A). Gene-concept analysis using these data showed a network in which both TNF and cytokine production share many genes with the other GO categories (Figure 5e; Appendix A). Visualization of the DEG in the TNF KEGG pathway (mmu04668) confirmed a higher abundance of Inact-up-DEG in all the steps of this signaling pathway that leads to leukocyte recruitment and activation (Appendix A). Hence, modulation of the immune response seems to be another relevant process during testis regression in this species.

## 4. Discussion

We studied the circannual reproductive cycle of the Mediterranean pine vole, *M. duodecimcostatus*. The distribution area of this species includes the south-eastern of France and most of the Iberian Peninsula. In northern populations reproduction takes place throughout the year, no resting period exists [46] as described also for the Lusitanian pine vole, *Microtus lusitanicus* [47], another Western European endemic sister species, which diverged from *M. duodecimcostatus* around 150 kyr ago [48]. In contrast, in the populations we studied here, which are at the meridional limit of their distribution area, Mediterranean pine voles may exhibit different breeding patterns at different geographically close locations. In the wastelands, reproduction is halted during the summer months, whereas in poplar plantations reproduction continues throughout the year, as it happened with animals maintained in captivity. Since all three groups of males studied here during the summer (wastelands, poplar groves and animal house) were subjected to the same photoperiod, our observations strongly suggest that it is the microenvironment in which each vole population is living what determines its reproductive status and the local existence or not of seasonal breeding. In fact, the living conditions in the poplar groves during the summer are much better than those in the wastelands, where temperature may exceed 45 °C in July-August and the land becomes completely dry and hard, which oblige voles to remain underground most of the daytime and preclude them from digging new tunnels. Contrarily, in poplar groves sunshine never reach the ground during the summer, humidity is high (due to frequent irrigation), temperature is moderate (never higher than 30–35 °C), food is abundant and the soil is soft, permitting voles to renew their tunnel network also in this season. Similarly, the living conditions in the animal house were also optimal, including both food and water supply ad libitum and controlled temperature. So, when life conditions are favorable, either natural or artificial, voles reproduce continuously, whereas in poor environments reproduction is halted and testis regression occurs. These results show that Mediterranean pine voles represent a very clear example of photoperiod-independent opportunistic breeding. Similarly, other populations of small mammals studied at lower latitudes of the temperate zone of the Earth do not respond to variation in photoperiod, indicating that opportunistic reproduction is quite common among small mammals [49]. Unlike the photoperiod, the mechanisms of action of these environmental cues remain unknown, although numerous investigations are being done, mainly in canary birds [50] and hamsters [51]. In the latter, it was shown that a non-photic environmental cue, food availability, may influence the expression of Kisspeptin and RFRP, two hypothalamic neuropeptides known to be involved in the control of the HPG axis, if photoperiod is maintained at an intermediate rhythm (13.5 h light), but not at the stimulating long-day pattern (16 h light). Hence, non-photic cues are thought to modulate the timing of reproduction in photoperiod-dependent species, but little is known about their function in species in which photoperiod appears to play a less important role. This is the case of the Mediterranean pine vole populations we have investigated here, which could become a useful animal model for the study of the genetic and functional response to the environmental cues influencing seasonal reproduction.

We studied the seasonal changes occurring in the testes of five species of micromammals living in the area around the city of Granada, in the Southeastern Iberian Peninsula. In two of these sympatric species, the Algerian mouse, *Mus spretus* [52] and the greater white-toothed shrew, *Crocidura russula* [20], males do not undergo testis regression during the non-breeding season, which takes place in winter, probably due to the lack of female receptiveness in this period. Contrarily, in the other three species, the Iberian mole, *Talpa occidentalis* [9,10,53], the wood mouse, *Apodemus sylvaticus* [52], and the Mediterranean pine vole, *M. duodecimcostatus* (present paper), testis regression occurs (facultatively in the later species) during the non-reproductive period, which takes place in summer. Three of these species are rodents (*M. spretus*, *A. sylvaticus* (family Muridae) and *M. duodecimcostatus* (family Cricetidae)) whereas the other two are Eulipotyphlans (*T. occidentalis* (family Talpidae) and *C. russula* (family Soricidae)). Thus, phylogeny seems not to be associated with the existence or not of seasonal testis regression. Additionally, even populations of the same species may show latitudinal differences in their seasonal reproductive patterns (summer breeding in the north and winter breeding in the south), showing that seasonal breeding is not a species-specific but rather a population-specific feature [20]. The lack of evident causes that may explain this surprising divergence between closely related species and even between northern and southern populations of the same species clearly argues that the mechanisms regulating both seasonal breeding and testis regression may evolve very rapidly, permitting this way particular populations to adapt to specific local environmental conditions. If this were the case, then males from northern and from southern populations of these species would have acquired some genetic differences responsible for their different circannual reproductive pattern. However, here we show that males captured in two locations separated by only 9 km, which are expected to have no genetic divergence with each other, may also exhibit completely different reproductive patterns. Hence, the most plausible explanation is that, at least in opportunistic breeders, the genetic and physiologic mechanisms controlling reproduction are in fact very plastic, allowing every particular individual (not population or species) to respond properly to heterogeneous and/or changing environmental conditions. Despite this, males of all aforementioned species exhibit reduced levels of circulating testosterone during the non-breeding season, indicating that the modulation of the molecular mechanisms regulated by this hormone is a common factor involved in testis regression. Reduction of testosterone levels is associated with spermatogenesis interruption and disruption of the BTB [54,55], two features that we have observed in species with regressed testes during the non-breeding season [2]. However, divergence also exists when we observe the spermatogenic status of the regressed testis in different species. In *T. occidentalis* and *A. sylvaticus,* meiosis initiation is not halted, so that early spermatocytes are present even in fully regressed testes. However, meiotic cells do not progress beyond the zygotene-pachytene stage because they are eliminated by apoptosis [9,10,52]. In contrast, here we show that spermatogonia do not enter meiosis at any time of the non-breeding period of *M. duodecimcostatus*, and regressed seminiferous tubules are composed only of Sertoli and spermatogonial cells. Accordingly, no apoptosis is needed in these testes as there are no spermatocytes to be depleted.

All these species-specific differences make it very difficult to establish a general mechanism to explain testicular regression in mammalian species. In this context, the study of the differences at the gene expression level between active and inactive testes in different species will help to explain which mechanisms and gene pathways are conserved, and which ones are species-specific. Here we report the first comprehensive study of the transcriptomes of active and inactive testes in a mammalian species, and have identified several biological processes and signaling pathways that are altered during testis regression. As expected, we found that genes involved in promoting spermatogenesis are preferentially upregulated in the active testes (Figure 5c; see GO:0007283 in Appendix A). Many of them are expressed in germ cells and, as mentioned above, this is probably an apparent upregulation as a consequence of the different amount of germ cells that both types of gonads contain. However, we also found Sertoli cell markers involved in the regulation of spermatogenesis, including SOX9 and SOX8 [56], WT1 [57] and GATA4 [58], that were upregulated in the inactive testes. We showed that SOX9 expression losses its spermatogenic cycle-dependent pattern in the inactive testis of *M. duodecimcostatus*, as it also happens in the inactive testes of the Iberian mole, *T. occidentalis*, where it is upregulated as well [37]. Since this gene is involved in the regulation of the expression of cell adhesion molecules [56], this transcription factor may have an important role in the Sertoli cell control of spermatogenesis during seasonal breeding by regulating the expression of molecules involved in BTB formation and function.

We found that the MAPK/ERK1/2, WNT, TGF-β, Cytosolic Ca2+ and PI3K signaling pathways are deregulated in the inactive testis of *M. duodecimcostatus*. All these pathways operate in Sertoli cells and are involved in the regulation of spermatogenesis and the dynamics of tight and adherens junctions present in the BTB [42,43,54,59,60,61,62,63,64,65,66]. Our analysis showed that MAPK/ERK1/2, a signaling pathway also required for mitotic cell proliferation and meiosis [60,62,66], plays a central role in this process. We found that 89 genes belonging to the MAPK/ERK1/2 signaling pathway were upregulated in the inactive testis (see GO:0043410 in Appendix A), and that many of these genes are also shared by other deregulated pathways (Figure 5d; Appendix A), indicating that all these signaling pathways probably play important roles in the Sertoli cell regulation of gamete production during seasonal breeding. Previous studies have shown that testosterone modulates the activity of the MAPK/ERK signaling pathway in Sertoli cells through a non-classical androgen receptor (AR) pathway, as shown by the activation of this pathway both in a testosterone suppression model resulting in germ cell depletion [60], and in rat cultured Sertoli cells stimulated with testosterone [67,68]. In addition, testosterone can also modulate the activation of other pathways such as the cytosolic Ca2+ concentration in Sertoli cells [69]. Altogether, these data indicate that the decrease of serum testosterone levels in the inactive males of *M. duodecimcostatus* affects the regulation by Sertoli cells of several interconnected molecular pathways that lead to (1) alterations of the spermatogenic cycle, (2) deregulation of cell–cell adhesion molecules and (3) disruption of the BTB dynamics. Concluding, our results suggest that the MAPK/ERK signaling pathway plays a central role as a mediator between testosterone signaling and Sertoli cell function.

The testis is subjected to a special immunological environment known as “immune privilege” that protect germ cells from autoimmune attack. This “immune privilege” is based on (a) the formation of the BTB in the seminiferous epithelium, (b) the diminished capability of the testicular macrophage population to mount an inflammatory response and (c) the constitutive expression of anti-inflammatory cytokines by immune and other somatic cells [70,71,72]. Under normal physiological conditions, testicular macrophages present relatively low inflammatory responses and high immunosuppressive properties compared with the macrophages from other tissues as shown by a reduced capability to synthesize IL-1β and TNF-α [73,74]. Our transcriptomic analysis indicated that the macrophage population was activated in the regressed testis of *M. duodecimcostatus* (Figure 5e; Appendix A and Appendix A). In addition to macrophages, T lymphocytes were also found in the interstitial spaces of the rat testis, including natural killer (NK) T cells and CD4+CD25+ regulatory T cells (Tregs). This latter cell population has a powerful immunosuppressive action that contributes to the tolerogenic environment of the testis [75]. Our analysis showed that T-cells were also activated in the regressed testis of *M. duodecimcostatus* (Appendix A). Thus, if we consider these results together with the observations that (1) testosterone levels were reduced in sexually inactive males of *M. duodecimcostatus*, (2) testosterone induced a reduction of proinflammatory cytokines in macrophages (including TNF-α) [76], (3) genes belonging to the TNF pathway and cytokine secretion were upregulated in the inactive testes of *M. duodecimcostatus* (Figure 5e; Appendix A and Appendix A), (4) TNF signaling inhibited the immune-suppressive function of Tregs [77] and (5) testosterone can induce an increase in the number of Tregs in the testes [78], we can conclude that the reduced levels of testosterone in the inactive testes of *M. duodecimcostatus* may induce the secretion of proinflammatory cytokines by macrophages and, probably, by other somatic cell types (e.g., Sertoli and peritubular myoid cells) that activate the macrophages and diminish the Tregs population. In a similar study performed in European beavers, the authors also reported altered inflammatory processes in the testes of non-breeding males [79]. These facts, together with the BTB disruption that takes place in regressed testes, will probably lead to the abrogation of the “immune privilege” that operates in active testes, which is necessary for the maintenance of spermatogenesis and male fertility [70,72]. Our results evidence that the immune system plays an important role during the testicular regression of seasonal breeders, although this function has hardly been studied during this process. Therefore, future studies aimed to unravel how the different components of the immune system vary during the testicular regression of different seasonal breeders will help us to understand its role in this process.

## 5. Conclusions

The Mediterranean pine vole, *M. duodecimcostatus,* represents a paradigmatic example of opportunistic breeding mammal as only males living in the wastelands undergo testis regression during the summer, whereas those living either in close poplar plantations of the same region or in our animal house keep breeding throughout the year. Unlike other mammals, in the regressed testes of these voles spermatogenesis was halted at the spermatogonial stage, existing no cells completing the initial stages of the meiotic prophase. During testis regression, the BTB becomes permeable, serum testosterone levels were reduced but testicular was is not increased. Finally, our transcriptome analysis indicate that both the suppression of the testis “immune privilege” and the deregulation of the MAPK/ERK signaling pathway probably play important roles in mammalian seasonal testis regression.

## Figures and Tables

**Figure 1 animals-11-01639-f001:**
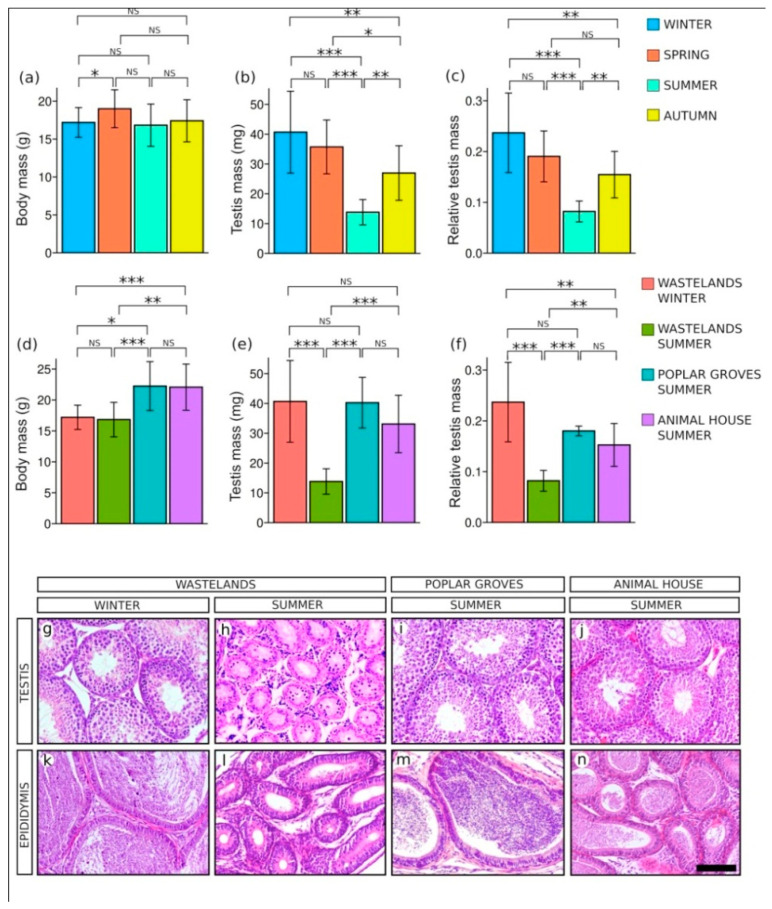
Morphometric analysis (**a**–**f**) and histology of testes and epididymes (**g**–**n**) of *M. duodecimcostatus* during the seasonal reproductive cycle. (**a**–**c**) Comparisons of (**a**) body mass, (**b**) testis mass and (**c**) relative mass of the testis of voles captured in the wastelands during the seasonal breeding cycle. (**d**–**f**) Comparisons of (**d**) body mass, (**e**) testis mass and (**e**) relative mass of the testis of voles living in different habitats. (**g**–**n**) Hematoxylin and eosin-stained histological sections of testes (**g**–**j**) and epididymides (**k**–**n**) of voles captured in the wastelands in winter (**g**,**k**), captured in the wastelands in summer (**h**,**l**), captured in poplar groves in summer (**i**–**m**) and living in our animal house facility during the summer (**j**,**n**). Scale bar shown in (**n**) represents 100 μm for all pictures. NS, not significant; *, *p* < 0.05; **, *p* < 0.005; ***, *p* < 0.001.

**Figure 2 animals-11-01639-f002:**
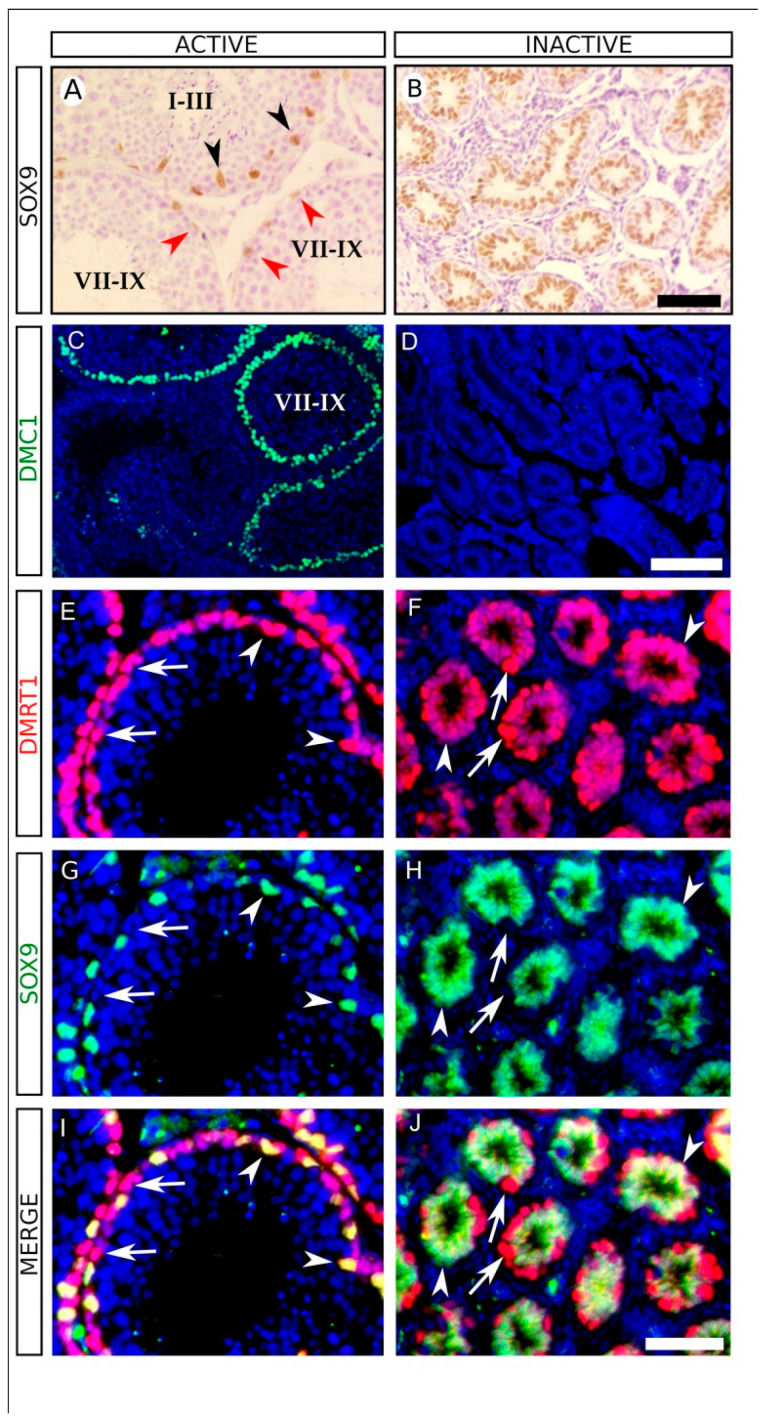
Immunohistological analysis for several Sertoli and germ cell-specific molecular markers in testis sections from *M. duodecimcostatus* captured in the wastelands. (**A**,**B**) Immunohistochemistry for SOX9. In winter (**A**) Sertoli cell expression of SOX9 is stronger in stages I-IV (black arrowheads) and weaker in stages VII-X (red arrowheads). However, in summer (**B**) SOX9 expression is strong and uniform in all Sertoli cells. (**C**,**D**) Immunofluorescence for DMC1. In winter (**C**) DMC1 is expressed in testis tubules in the spermatogenic-stages VII-IX. In contrast, no expression for DMC1 was observed in the summer (**D**). (**E**–**J**) Double immunofluorescence for DMRT1 (red) and SOX9 (green). In winter, DMRT1 is expressed in spermatogonial cells (arrows) and in Sertoli cells (arrowheads), where it colocalizes with SOX9. Note that, inside the tubules, there are many SOX9^−^ DMRT1^-^ cells corresponding to germ cells in more advanced stages of the spermatogenic cycle (**E**,**G**,**I**). In summer, SOX9^+^, DMRT1^+^ Sertoli cells fill almost the entire inner part of the tubules (arrowheads), and SOX9^−^, DMRT1^+^ spermatogonial cells located at basal positions (arrows) can also be observed (**F**,**H**,**J**). No SOX9^−^, DMRT1^-^ cells was observed inside of the tubules (**F**,**H**,**J**). Testes sections in (**C**–**J**) were counterstained with DAPI. Scale bar shown in (**B**) represents 50 μm for (**A**,**B**). Scale bar shown in (**D**) represents 100 μm for (**C**,**D**). Scale bar shown in (**J**) represents 50 μm for (**E**–**J**).

**Figure 3 animals-11-01639-f003:**
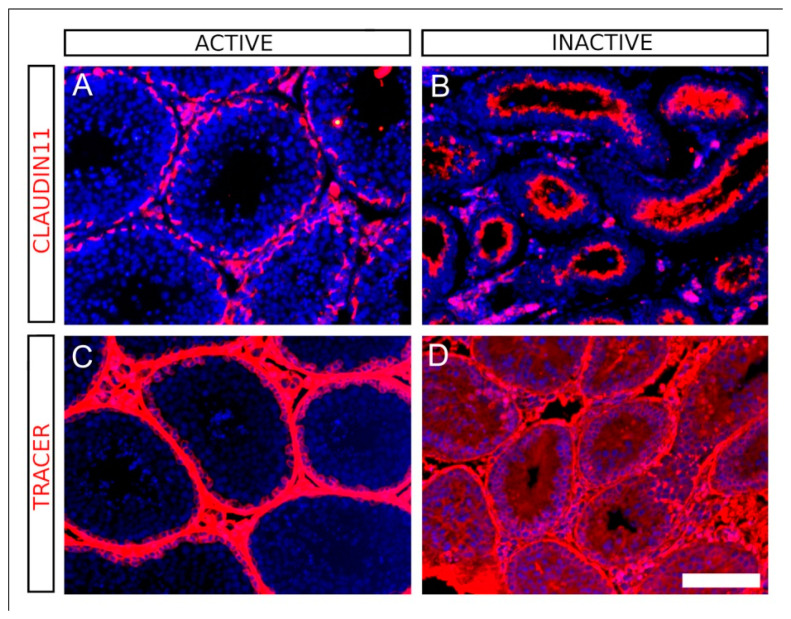
The BTB of regressed testes of *M. duodecimcostatus* is impaired. (**A**,**B**) Immunofluorescence for CLAUDIN11 in testes of voles captured in the wastelands. In winter (**A**), a sharp CLAUDIN11 immunoreactivity surrounding the seminiferous tubules just at the limit between the basal and the adluminal compartments can be neatly observed. In contrast, CLAUDIN11 expression pattern appears completely disorganized in the summer (**B**). (**C**,**D**) Test of BTB functionality using a biotin tracer (red fluorescence) of voles captured in the wastelands. In winter (**C**), the tracer did not penetrate beyond the basal compartment, whereas in summer immunoreactivity was detected in deepest areas of the regressing seminiferous tubules (**D**). Testis sections were counterstained with DAPI. Scale bar shown in (**D**) represents 100 μm for all pictures.

**Figure 4 animals-11-01639-f004:**
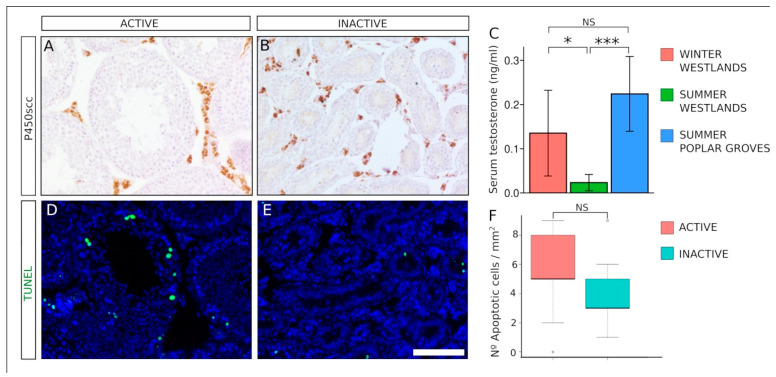
Reduced steroidogenesis but not increased apoptosis in regressed testes of *M. duodecimcostatus* (**A**,**B**). Immunohistochemistry for cholesterol side-chain cleavage enzyme (P450scc) in testis sections of voles captured in the wastelands showed a strong expression in Leydig cells with similar signal intensity during both winter (**A**) and summer (**B**). (**C**) Serum testosterone concentrations of male voles captured in different habitats and seasons. (**D**,**E**) TUNEL assay in testis sections of voles captured in the wastelands showed that in both winter (**D**) and summer (**E**) the number of apoptotic cells was very reduced. (**F**) Quantification of the number of apoptotic cells in winter and summer testes. Testes sections in (**D**,**E**) were counterstained with DAPI. Scale bar shown in (**D**) represents 100 μm for all the pictures. NS, not significant; *, *p* < 0.05; ***, *p* < 0.001.

**Figure 5 animals-11-01639-f005:**
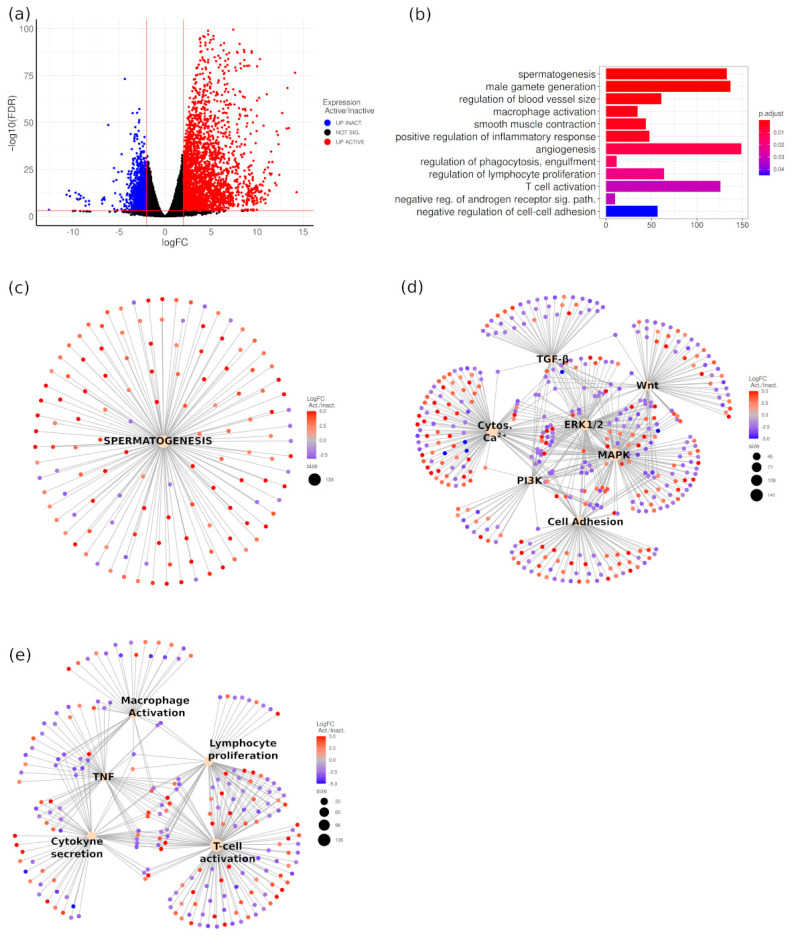
Transcriptomic analysis of active and regressed testes of *M. duodecimcostatus*. (**a**) Volcano plot of the differential expression test. (**b**) Gene Ontology analysis of the deregulated genes revealed a significant enrichment (p.adjust< 0.05) in terms associated to normal testicular functions. (**c**) Cnetplot of the GO term spermatogenesis. Note that most of the genes are upregulated. The names of the genes are depicted in electronic Appendix A. (**d**) Cnetplot of several significantly enriched GO terms of molecular pathways identified in our Gene Ontology analysis. The names of the genes are depicted in electronic Appendix A. (**e**) Cnetplot of several significantly enriched GO terms associated with the activation of the immune system. The names of the genes are depicted in electronic Appendix A. In pictures (**a**,**c**–**e**) red color indicates upregulation in the active testis and blue color upregulation in the regressed testis. In figures (**d**,**e**), the size of sepia circles is proportional to the number of deregulated genes they represent.

## Data Availability

The transcriptome data presented in this study are openly available in ArrayExpress, accession E-MTAB-10291.

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
