# Peer review of "Mediterranean Pine Vole, Microtus duodecimcostatus: A Paradigm of an Opportunistic Breeder"

_animals, 2021, doi:10.3390/ani11061639_

Round 1

Reviewer 1 Report

This is a quite interesting paper dealing with the environmental effects on reproductive seasonality of the male pine vole. The paper is well written, the hypothesis is well and thoroughly investigated, thus, this reviewer believes that the paper must gain publication priority. However, some issues should be sorted out before final acceptance.

  1. As long periods post-capture could have induced stress-related reactions, please indicate what the time spell between capture and examinations on the wild voles was.
  2. Authors conclude that environmental conditions-mainly food availability- drive the observed differences between the two types of animals. It would be very interesting this hypothesis to be -somehow- supported by data. As for example, fat deposition, levels of metabolic signals (glucose, insulin, NEFA, leptin etc)
  3. L202 -203 Why did you remove germ-cell specific genes? Maybe there were differences between active and inactive genes. how many genes were removed? Was their representation equal in both libraries?
  4. L377-379 Did you try to normalize the expression without removing genes? What were the results?
  5. L563 -568 As it seems, genes may show pleiotropy, so, by removing specific genes you might have lost useful information about their roles in regulating other pathways

Minor

  1. The title needs rephrasing: to my understanding Paradigm and Example are synonymous. It would be advisable: a paradigm of an opportunistic breeder, to be used
  2. Lines 25-26 please rephrase
  3. L 67-69 please delete the last sentence of this paragraph
  4. L63-94 this sentence needs rephrasing
  5. 116 please define what immune privilege is
  6. L382-384 What are the correlation coefficients in the groups?
  7. L388-389 It was not just "some genes"; according to Fig 5 b,  they are highly enriched in GO Terms

Author Response

Reviewer 1

We thank the two reviewers for their constructive criticisms to our manuscipt, that helped us to deliver a quite improved version of it.

1. Comment: As long periods post-capture could have induced stress-related reactions, please indicate what the time spell between capture and examinations on the wild voles was.

Response. The reviewer is right and we agree that this information must be included in the Material and Methods section. Accordigly, we modiffied the text in lines 150-151 as follows:

Animals were transported to our laboratory and euthanised by CO2 inhalation the day of capture, dissected and….”

2. Comment: Authors conclude that environmental conditions-mainly food availability- drive the observed differences between the two types of animals. It would be very interesting this hypothesis to be -somehow- supported by data. As for example, fat deposition, levels of metabolic signals (glucose, insulin, NEFA, leptin etc).

Response. The reviewer sugestion is very good and could support the hypothesis that food availability is a key cue controlling the reproductive rythm of this species. However, we are afriad it is currently impossible for us to perform these experiments, as we do not have the required material and would need to capture a good number of new animals, both in summer and winter. Nevertheless, since such a study, by itself, could serve to prepare a new, independent manuscript, we will keep it in mind for potential future reseach.

3. Comment: L202 -203 Why did you remove germ-cell specific genes? Maybe there were differences between active and inactive genes. how many genes were removed? Was their representation equal in both libraries?

Response. As we show in Figure 2, active testes contain germ cells in all the spermatogenic stages, whereas inactive testes contain only spermatogonia. Thus, in the inactive gonad, genes involved in meiosis spermatid differentiation and sperm maturation are not expressed. Initially we analysed the transcriptomic data without removing germ cell-specific counts. In this case our bioinformatic analyses mainly revealed, in active testes, an enrichment in genes and GO terms related to meiosis, spermatogenesis and spermiogenesis. However, this enrichment did not reflect differences in gene regulation between the two conditions, but the absence of germ cell types in the inactive testes. These results could be predicted by observing the histology of active and inactive testes, but will not help us understand the molecular mechanisms operating in the regressed testis.

In addition, for our transcriptomic analyses we must consider that a) somatic cell function is essential for testis regression and b) differential expression analysis of transcriptomic data rely on the number of reads mapped to a gene / total number of reads (e.g. tmm, cpm, rpkm...etc). Thus, if we perform a bioinformatic analysis in which we include a high number of germ cell-specific transcript reads in the active testes but not in the inactive testes, the somatic-specific transcript counts in the active testes will be underestimated when compared with the inactive testes. Ideally, for this analysis, the germ cell transcript reads should be removed. Although this is not possible when doing whole testis transcriptome, we could perform a bioinformatic approach using the recently published single cell transcriptome of mouse adult testes (Hermann and Green). In these studies the authors provide gene expression signatures of every cell type, including spermatocytes and posterior spermatogenic stages, that we used to remove germ cell-specific transcripts.

From the original assembled transcriptome fasta file, we discarded 54618 transcripts corresponding to germ cell signature genes and we keeped 59351 transcripts.

As we mention in supplemental methods, to assess the effect of this normalization method we calculated the correlation coefficients between all possible sample pairs. The main difference between active and inactive testes is the presence of germ cells in the active ones. After the removal of the transcripts from germ cells signatures genes we expect a smaller difference between these coefficients as samples from different category must become more similar, and accordingly, the mean correlation coefficient between samples from different category increased from ~0.5 to ~0.7. Furthermore, when removing the noise caused for this differences in cell composition the real signal became more clear as we identified gene pathways and biological processes operating in somatic testicular cells that previous studies showed to be altered when testosterone levels are reduced in the testes, again proving that our approach has a strong biological significance.

4. Comment: L377-379 Did you try to normalize the expression without removing genes? What were the results?

Response. We already answered this question in the previous point.

5. Comment: L563 -568 As it seems, genes may show pleiotropy, so, by removing specific genes you might have lost useful information about their roles in regulating other pathways

Response. Of course genes may play a role in several cell types but if a gene is expressed in both germ and somatic cells, in a whole testis transcriptome it is not possible to know which transcripts belong to each of the cell types in which this gene is expressed. Thus, the data from this gene are not informative and only contribute the background noise. Removing transcripts from this genes does not necessarily means that we miss the whole pathway because others somatic specific genes in this same pathway may provide enough information to be detected in a GO enrichment analysis. Our study show that only after removing germ cells signature genes altered pathways that operates in somatic cells are revealed.

Minor comments

  1. Comment: The title needs rephrasing: to my understanding Paradigm and Example are synonymous. It would be advisable: a paradigm of an opportunistic breeder, to be used

  2. Response: We changed the title as suggested by the reviewer

  3. Comment: Lines 25-26 please rephrase.

    Response: Done as in the title.

  4. Comment: L 67-69 please delete the last sentence of this paragraph

    Response: We removed the last sentence of the paragraph

  5. Comment: L63-94 this sentence needs rephrasing

    Response: We understand the reviewer wanted to write L93-94. We have rephrased this sentence as follows:The seasonal breeding pattern of different populations of the same species can also vary with latitude, as described for instance...”

  6. Comment: L116 please define what immune privilege is

    Response: The term “immune privilege” is defined later in the text, so we have removed it from this last sentence of the Introduction section, which was modified as follows: “...deregulated, and the intensity of the immune response is altered.”

  7. Comment: L382-384 What are the correlation coefficients in the groups?

    Response: We consulted or colleagues for the Statistics Department and they said that calculating a correlation coefficient between two variables, when one of them has only two values (active and inactive, in this case) does not make much sense. Better, Student's t test can be used to test for differences of one variable (testosterone level) between the two categories of the other one (active vs inactive), which is what we did in our study. Nonetheless, we calculate the correlation coefficient: the value of r is 0.5935, and the P value is 0.015366. Thus, as expected, there is a correlation between the status of the testis (active or inactive) and the testosterone level, as there is a significant difference between the means of testosterone level between these two categories according to the Student's t-test. In our opinion, this data is not relevant for the study, but we could include it if the reviewer consider it is necessary.

  8. Comment: L388-389 It was not just "some genes"; according to Fig 5 b, they are highly enriched in GO Terms

  9. Response: The reviewer is right. We susbtituted “some” with “many” in that sentence.

Reviewer 2 Report

In the manuscript titled “Mediterranean pine vole, Microtus duodecimcostatus:  para- digmatic example of opportunistic breeder” the authors studied the circannual reproductive cycle in males of the Mediterranean pine vole, Microtus duodecimcostatus, in the south-eastern Iberian Peninsula, performing morphological, hormonal, functional, molecular and transcriptomic analyses and demonstrating a para-digmatic example of opportunistic breeder.

In my opinion, this work is very interesting and deserves to be published in Animals but before recommending the publication in Animals it is necessary to provide further details. Therefore I recommend a major revision.

My observations are the following:

Simple Summary needs revision of English form

Lines 182 and 184: CaCl2 2 must be subscript

Line 358 correct ml to mL and also in all other parts of the manuscript

Line 365 “incidence of cell death in both active and inactive the testes of M. duodecimcostatus males” delete the before testes

Line 369 “5,6 ± 2,3 apoptotic cells / 0,1 mm2 of testis section; inactive : 4.0 ± 2,3 cells/0.1 mm2… ……………..mm2      2 should be superscript

In literature there are some works regarding the seasonal dependence of molecular effects of some heavy metals such as cadmium on the properties of the proteins which organize sperm chromatin Mytilus galloprovincialis . In some organisms, in the non-reproductive period these proteins do not bind DNA and this affects several aspects. In addition the pollutants are accumulated in the sperm only in winter and not in summer. I think that the authors should consult these papers and mention this in the discussion to argue about any differences found during the different seasonal periods they tested.

In Mytilus galloprovincialis it is also reported that under sever hyposaline conditions, phenomena of fertility preservation strategy occur due to gamete plasticity. The authors should also mention these examples in the discussion.

The authors state that a significant difference in the testes in the different periods of the year was found. I think that it is necessary to explain why even better so that all readers can understand.

Pollution has a strong impact on the reproductive sphere of many organisms.  The differences in the type of pollutants found in this work must be highlighted because the different pollutants can produce different effects in the reproductive system.

Spermatozoa are now considered early sentinels of the health of the environment and organisms. I therefore believe that the work should emphasise how readily the reproductive system responds to environmental changes.

The work should also mention men living in polluted environments and how many changes, at a molecular level, occur in their spermatozoa. It has been shown that there are changes in the classical parameters of spermatozoa as well as in molecular and biochemical parameters.

In this regard, I recommend reading and citing some of the works and seeing if there are any links with their results.

  • For this aim I suggest to read and eventually quote the following works:

doi: 10.3390/ijms21124198

doi: 10.3390/ijms21186710

Author Response

We thank the two reviewers for their constructive criticisms to our manuscipt, that helped us to deliver a quite improved version of it.

Comment: Simple Summary needs revision of English form

Response: Several changes have been performed in the Simple Summary in order to make ir clearer for a non-specialized reader (see the revisd version of the manuscript).

Comment: Lines 182 and 184: CaCl2 2 must be subscript

Response: Done (line 194 of the revised version)

Comment: Line 358 correct ml to mL and also in all other parts of the manuscript

Response: The change was made throughout the text a total of 7 times.

Comment: Line 365 “incidence of cell death in both active and inactive the testes of M. duodecimcostatus males” delete the before testes

Response: Deleted (line 382 of the revised version)

Comment: Line 369 “5,6 ± 2,3 apoptotic cells / 0,1 mm2 of testis section; inactive : 4.0 ± 2,3 cells/0.1 mm2… ……………..mm2 should be superscript

Response: Done (line 387 of the revised version)

Comment: In literature there are some works regarding the seasonal dependence of molecular effects of some heavy metals such as cadmium on the properties of the proteins which organize sperm chromatin Mytilus galloprovincialis. In some organisms, in the non-reproductive period these proteins do not bind DNA and this affects several aspects. In addition the pollutants are accumulated in the sperm only in winter and not in summer. I think that the authors should consult these papers and mention this in the discussion to argue about any differences found during the different seasonal periods they tested. In Mytilus galloprovincialis it is also reported that under sever hyposaline conditions, phenomena of fertility preservation strategy occur due to gamete plasticity. The authors should also mention these examples in the discussion.

Response: According to the reviewer suggestions, we consulted several recent articles (DOI: 10.1016/j.crvi.2014.05.003 and DOI: 10.1002/mrd.23240 , for instance). In our opinion, conducting new reaserch on the possible involvement of heavey metals, like cadmium, in the seasonal reproductive pattern of small mammals, like Microtidae, would be interesting, but the conceptual connection between the topics addressed in these papers and those of our present manuscript is very weak. Regarding the capacity to accumulate heavey metals, small mammals and mussels are not comparable as the later are filtering marine animals living in the coasts. Our manuscript is focussed on the causes of seasonal reproduction of a mammalian rodent (both photoperiod-dependent and -independent), whereas these papers are focussed on the seasonal alterations in the concetration of these metals in the mussel tissues, including the reproductive tract. The fact that these metals may affect sperm functionality with a seasonal pattern is just a circumstancial relationship with mammalian seasonal breeding and seasonal testis involution. So, we do not see how to establish a reasonable relationship between the two stories.

Comment: The authors state that a significant difference in the testes in the different periods of the year was found. I think that it is necessary to explain why even better so that all readers can understand.

Response: Without more specific indications, we do not see how “to explain why even better”. In fact, the entire discussion section is dedicated to do it. Please, let us know specifically where a better/longer explanation is needed.

Comment: Pollution has a strong impact on the reproductive sphere of many organisms. The differences in the type of pollutants found in this work must be highlighted because the different pollutants can produce different effects in the reproductive system. Spermatozoa are now considered early sentinels of the health of the environment and organisms. I therefore believe that the work should emphasise how readily the reproductive system responds to environmental changes. The work should also mention men living in polluted environments and how many changes, at a molecular level, occur in their spermatozoa. It has been shown that there are changes in the classical parameters of spermatozoa as well as in molecular and biochemical parameters. In this regard, I recommend reading and citing some of the works and seeing if there are any links with their results. For this aim I suggest to read and eventually quote the following works: doi: 10.3390/ijms21124198; doi: 10.3390/ijms21186710

Response: We agree with the reviewer in the importance of studies devoted to unravel the growing incidence that pollutans have on animal reproductive performance, and we even reiterate that new reaserch on the possible involvement of pollutants in the reproduction performance of small mammals is interesting of course, but the topics the reviewer mention are only barely related to the contents of our current manuscript. These papers deal with the effects of certain pollutants on DNA oxidative damage and how this affect sperm features and performance, a general effect that has little connection with the causes of seasonal testis regression. Whereas the seasonal testis regression we describe in voles implies the complete loss of the germinative epithelium, the men living in polluted environments studied in one of these papers, instead, underwent some sperm alterations but maintained their germinative epithelium and showed no seasonal impact on their fertility. Hence, again, we do not see how to establish a reasonable relationship between the two topics.

Round 2

Reviewer 2 Report

Accept in present form